# COVID-19 Vaccine Hesitancy among Adolescents: Cross-Sectional School Survey in Four Chinese Cities Prior to Vaccine Availability

**DOI:** 10.3390/vaccines10030452

**Published:** 2022-03-15

**Authors:** Palizhati Rehati, Nubiya Amaerjiang, Liping Yang, Huidi Xiao, Menglong Li, Jiawulan Zunong, Long Wang, Sten H. Vermund, Yifei Hu

**Affiliations:** 1Department of Child, Adolescent Health and Maternal Care, School of Public Health, Capital Medical University, Beijing 100069, China; plzt1998@163.com (P.R.); 13693617970@163.com (N.A.); 13426483171@163.com (L.Y.); xhd19988023@163.com (H.X.); limenglong_ph@163.com (M.L.); jiawulanzn@ccmu.edu.cn (J.Z.); wanglong201506@sina.com (L.W.); 2Department of Epidemiology of Microbial Diseases, Yale School of Public Health, Yale University, New Haven, CT 06510-3201, USA; sten.vermund@yale.edu

**Keywords:** vaccination, vaccine hesitancy, adolescents, willingness, COVID-19 vaccine

## Abstract

To address the novel coronavirus disease (COVID-19) pandemic, development and regulatory evaluations have been accelerated for vaccines, authorizing emergency use. To anticipate vaccine preparedness in adolescents, we studied COVID-19 vaccination awareness and willingness to vaccinate before the vaccine became available. We conducted a cross-sectional survey among 9153 (4575 boys, 50%) students with a mean age of 14.2 years old in four cities in China to collect information on demographic characteristics and their COVID-19 vaccination concerns. Multinomial logistic regression was used to analyze the influencing factors of vaccine hesitancy (“not sure”) and resistance (“do not want it”). The results showed that 2891 (31.6%) were hesitant and 765 (8.4%) were resistant to being vaccinated. Additionally, multivariable analyses showed that vaccine hesitancy and vaccine resistance were associated with living in the Beijing area (OR = 1.62; 95% CI: 1.40–1.88; OR = 1.81; 95% CI: 1.44–2.28), lack of influenza vaccination experience (OR = 1.33; 95% CI: 1.14–1.55; OR = 1.57; 95% CI: 1.25–1.98), no perceived susceptibility (OR = 1.72; 95% CI: 1.50–1.97; OR = 3.57; 95% CI: 2.86–4.46), and perceiving no cues to action (OR = 3.24; 95% CI: 2.56–4.11; OR = 27.68; 95% CI: 21.81–35.13). Postulating a highly effective vaccine (OR = 0.84; 95% CI: 0.72–0.98; OR = 0.66; 95% CI: 0.52–0.83) decreased both vaccine hesitancy and resistance. Vaccine hesitancy alone was associated with girls (OR = 1.21; 95% CI: 1.09–1.36) and was less common among students boarding at school (OR = 0.79; 95% CI: 0.68–0.92), postulating convenient vaccine access (OR = 0.84; 95% CI: 0.73–0.96), and having doctors’ recommendation (OR = 0.86; 95% CI: 0.76–0.98). In conclusion, the results of the study showed that vaccine hesitancy among students in China was associated with limited health literacy and lower risk awareness. Our findings in China suggest that educating youth regarding COVID-19 and the safety and effectiveness of immunization help reduce concerns and increase vaccine confidence and acceptance.

## 1. Introduction

Severe acute respiratory syndrome coronavirus-2 (SARS-CoV-2) has had a devastating global disease burden since its outbreak, with a wide range of clinical presentations from asymptomatic to fulminant disease [1]. As of late February 2022, the cumulative number of confirmed cases of COVID-19 and deaths exceeded 430 million and 5 million worldwide, respectively. Over 10 billion doses of the vaccine have been administered but the fully vaccinated coverage was only 55% of the global population as of February 2022; therefore, vaccination and boost initiatives must continue [2].

Highly effective treatments to lower the morbidity and mortality of COVID-19 remain elusive [3]. In contrast, the vaccination has proven both effective and affordable, such that regulatory efforts have been accelerated to authorize earlier emergency use. Once a vaccination was developed which benefited adults and only caused serious adverse consequences on rare occasions, studies of youths ages 12 and up were completed, demonstrating safety and immunogenicity. As of February 2022, pediatric trials of COVID-19 vaccines in many countries have nearly been completed [4,5,6]. Along with the issues of efficacy and the delivery of the vaccine, it is important to monitor public confidence in immunization programs. Vaccine hesitancy is a major barrier to vaccine uptake and achievement of herd immunity, which is critical to protect the most vulnerable populations and contain the pandemic [7].

The social psychology of vaccine hesitancy suggests that education and socio-economic status do not influence hesitancy in only one direction, but rather can be associated with both lower and higher levels of vaccine acceptance in differing contexts [8]. To better understand the role of the multiple factors influencing the acceptance of vaccines, we adopted the health belief model as a theory-based research framework to assess vaccination intentions among adolescents [9,10,11]. The health belief model includes several components: perceived susceptibility, severity of the disease, vaccine benefits, vaccine risks, barriers to access, self-efficacy, and perceived cues to action [12]. Other salient issues can include demographic characteristics, understanding of vaccine effectiveness and safety, and convenience and the price of the vaccine [13]. Previous studies on the willingness to receive the COVID-19 vaccination included adults [14,15] and university students [16,17,18]. The few studies which investigated vaccine hesitancy in adolescents [19,20,21] mainly focused on the influence of parents on adolescents’ willingness to vaccinate, while studies among Chinese adolescents were mostly conducted in one city.

Since people of all ages are susceptible to SARS-CoV-2, and given the risk of transmission from children to vulnerable people, the society strategy for herd immunity must include a COVID-19 vaccine plan for children and adolescents. Students in China are the bridge population, connecting families to school gatherings, and vice versa; large gatherings can amplify SARS-CoV-2 transmission. An increase in vaccination rates among children and adolescents is crucial. At the time of our study (December 2021), vaccination was not recommended for adolescents or children due to a lack of phase Ш clinical data, including dosing for people under 18 years old. Additionally, China commenced a COVID-19 vaccination plan for children and adolescents aged 3 to 17 in July 2021 [22].

To our knowledge, no large-scale surveys of the COVID-19 vaccination of children and adolescents are being conducted preceding the immunization plan implemented in China. Additionally, we conducted a large cross-sectional study of vaccine hesitancy in four cities in eastern, southern, central and northwestern China to study a representative cross-section of the children and adolescents in China.

We aimed to investigate willingness to be vaccinated among Chinese children and adolescents and to assess possible factors of vaccine hesitancy and resistance in order to focus on children and adolescents in the current vaccination process to prepare for the expansion of immunization coverage in China.

## 2. Materials and Methods

### 2.1. Study Design

We conducted a cross-sectional survey on the willingness to receive COVID-19 vaccines among junior high (middle) school and senior high (upper) school students in four Chinese cities (Beijing, Xi’an in Shaanxi province, Shenzhen in Guangdong province, and Anqing in Anhui province) from 8 to 30 December 2020.

We used the Wenjuanxing^®^ (Changsha Haoxing Information Technology Co. Ltd., Changsha, China) online survey tool (similar to SurveyMonkey^®^, San Mateo, CA, USA) for the informed consent process and administration of the online survey. We approached 9463 youth in grades 7–12 to answer the questionnaire, of whom 212 declined to participate (97.8% participation). After excluding invalid responses (inconsistent response across question options or plausibility of data ranges), 9153 questionnaires were finally included in the analysis. Figure 1 shows the participants’ recruitment and selection process.

Influenza immunization plans differed between Beijing and the other three cities, so we performed categorical analyses, comparing Beijing and the other cities for their demographic information. Schoolchildren in grades 7–12 were eligible in the four cities if they could read and comprehend the survey content and if they could independently complete the survey. We did not recruit adolescents from vocational schools.

We defined vaccine hesitancy as being unsure as to whether to accept the vaccination. Unwillingness to accept the vaccination was defined as vaccine resistance.

We calculated the sample size using the formula as *n* = *deff* × Z^2^_α_/_2_ × *p* (1 − *p*)/δ^2^ (where *n* is the sample size, *deff* is the design effect, we assigned *deff* = 2.0 for cluster sampling, Z is distribution function for two-tailed alpha = 0.05, *p* is the targeted prevalence using influenza vaccination rates as a reference, and δ is the acceptable standard error) [23] We chose *p* = 0.25 according to a meta-analysis of the influenza vaccination rate of people who were six months old to 17 years old from 2005 to 2017 estimates in China [24], assuming that the initial COVID-19 vaccination rates would be at least that of the influenza vaccination rate. With δ = 0.1*p*, we calculated the sample size as 2305 with 90% power to estimate the vaccine acceptance rate of the population. Assuming that 10–20% of participants might decline to participate in the survey, we planned a sample size of 2536–2767 respondents.

### 2.2. Measurement

The outcome variable was the willingness to receive the COVID-19 vaccination, once available, using the statement, “I am willing to get vaccinated”, and response options: “completely disagree”, “disagree”, “I don’t know”, “agree”, or “strongly agree”. For most analyses, we coded “completely disagree” and “disagree” into “resist to be vaccinated”, “I don’t know” as “hesitate to be vaccinated”, and merged “agree”, “strongly agree” into “willing”.

Covariates included sex, region, daily living, ethnicity, stage of schooling (junior (middle) or senior high (upper) school), and concerns about coronavirus and its vaccination, using statements such as, “I am afraid of coronavirus transmission”, “I have a risk of being infected by coronavirus”, and “the COVID-19 has made me pay more attention to the influence of infectious diseases in my life, influencing me to get/want an influenza vaccine”. We assessed the major concerns that youth report would affect vaccination decisions, including vaccine safety, effectiveness, price, convenience, and doctors’ recommendation. We included two items to explore a possible association of experiences of the influenza vaccination and attitudes towards the COVID-19 vaccination: “Have you been vaccinated for influenza this year?” and “I had a positive experience with my influenza vaccination”.

The health belief model (HBM) is a common tool to evaluate vaccine hesitancy. Based on available HBM questionnaires and scales [25,26,27,28], we developed the questionnaire to include sociodemographic characteristics. Briefly, HBM is comprised of six domains: perceived risk (perceived severity and perceived susceptibility, 2 items), perceived benefits and barriers (4 items) and cues to action (2 items), self-efficacy (2 items), demographic characteristics (5 items), and knowledge (1 item) (Figure 2). 

Most items used five-point Likert scales. After researchers’, teachers’, and students’ consultations, we revised the content and structure of the preliminary questionnaire and conducted pilot surveys (which was conducted among 84 junior high school students with ages of 15.9 ± 1.8 years old from 2 schools in Beijing, and 52.4% were boys) to test the reliability and validity. After four waves of modification, we developed the final questionnaire. And 16 items have been evaluated in this study.

### 2.3. Statistical Analysis

Participants’ characteristics were described using frequencies and percentages. Influencing factors of acceptance of a COVID-19 vaccine were assessed with Chi-square test, including variables with two-sided *p* ≤ 0.05 in multinomial logistic regression, presented with odds ratios (OR) and 95% confidence intervals (CI). All data were analyzed using Statistical Analysis System software (V.9.4, SAS Institute Inc., Cary, NC, USA).

## 3. Results

A total of 9153 respondents—ages ranging from 12.0 to 17.5 years, with an average age of 14.2 (SD = 1.6)—were finally included in the analysis.

### 3.1. Demographic Characteristics of the Participants

Among 9153 respondents, 765 (8.4%) were not willing to get vaccinated (resistant), 2891 (31.6%) were not sure about the vaccination, and 5497 (60.0%) were willing to receive a COVID-19 vaccine (Table 1).

### 3.2. Comparison of Vaccination Willingness

There was no difference between Han and ethnic minority youth in willingness to receive the vaccination (*p* = 0.84). In contrast, differences in vaccine hesitancy were noted by sex, daily living, stage of schooling, region, and perceptions of COVID-19 vaccination (Table 2).

### 3.3. Factors Influencing Vaccine Resistance

The likelihood ratio of multinomial logistic regression was 4288.2 (DF = 40, N = 9153, *p* < 0.001), and the values of Cox and Snell R^2^, and Nagelkerke R^2^ were 37.4% and 45.2%, respectively. The results showed that Xi’an students were more likely than the Beijing referent group to desire a COVID-19 vaccine (OR = 1.81; 95% CI: 1.44–2.28). Adolescents with no experience with the influenza vaccine (OR = 1.57; 95% CI: 1.25–1.98) or who were unsure about having it (OR = 1.30; 95% CI: 1.01–1.67) were more resistant to the COVID-19 vaccination (Table 3). The more effective the COVID-19 vaccine was believed to be, the less resistant students were to vaccinate (OR = 0.66; 95% CI: 0.52–0.83). Students who were not afraid of SARS-CoV-2 transmission were more vaccine-resistant (OR = 1.35; 95% CI: 1.07–1.71). Students who did not perceive that they were susceptible to COVID-19 (OR = 3.57; 95% CI: 2.86–4.46) or were unsure about the risk of infection (OR = 1.43; 95% CI: 1.08–1.89) were more vaccine-resistant. Students who thought that COVID-19 would have no impact on their lives (OR = 27.68; 95% CI: 21.81–35.13) or those who were not aware of any impact (OR = 8.03; 95% CI: 6.30–10.22) were more vaccine-resistant. No association of vaccine resistance was noted by sex, stage of schooling, daily living, receiving an influenza vaccine in 2020, concerns about a vaccine’s safety, price, convenience, or doctors’ recommendation (Table 3).

### 3.4. Factors Influencing Vaccine Hesitancy

Compared with boys, girls were more likely to be vaccine-hesitant (OR = 1.21; 95% CI: 1.09–1.36). The relative odds of vaccine hesitancy among students in Beijing were 1.62 times higher (OR = 1.62; 95% CI: 1.40–1.88) than that of school students in Xi’an. The odds of vaccine hesitancy among boarding students were 21% lower (OR = 0.79; 95% CI: 0.68–0.92) than non-boarders. Participants who had no experience with an influenza vaccine (OR = 1.33; 95% CI: 1.14–1.55) or were not sure about their experience (OR = 1.44; 95% CI: 1.24–1.67) were more likely to hesitate. The more effective the vaccine is believed, the less hesitant were the students to vaccinate (OR = 0.84; 95% CI: 0.72–0.98). Higher COVID-19 vaccine price concerns increased hesitancy (OR = 1.20; 95% CI: 1.05–1.36). Anticipated convenience to vaccinate reduced vaccine hesitancy (OR = 0.84; 95% CI: 0.73–0.96), as did doctors’ recommendation to vaccinate (OR = 0.86; 95% CI: 0.76–0.98). Students who did not know much about SARS-CoV-2 were more likely to be hesitant (OR = 2.09; 95% CI: 1.70–2.57), as were respondents who believed that COVID-19 would have little or no influence in their lives (OR = 3.24; 95% CI: 2.56–4.11) or were not concerned about COVID-19 (OR = 14.15; 95% CI: 12.27–16.31). Stage of schooling and having received an influenza vaccine in 2020 were not associated with COVID-19 vaccination hesitancy (Table 3).

## 4. Discussion

Our survey suggested that only 60% of school students in grades 7–12 wanted to receive a COVID-19 vaccine in China prior to the vaccine being available for them, in contrast to the much higher willingness indicated among Chinese adults during the active SARS-CoV-2 transmission period in Wuhan [15]. The students with concerns included 31.6% who were hesitant and 8.4% who were resistant to receiving a COVID-19 vaccine. Our findings were similar to other nations: in countries such as Sweden, Canada, the US, and England, around 30–40% of adolescents would hesitate to accept a COVID-19 vaccination [20,29,30,31].

There was a sex discrepancy between vaccine hesitancy, with girls less willing to receive a COVID-19 vaccine, consistent with previous studies [32,33,34]. We speculate that this may be due to misinformation circulating on social media about the COVID-19 vaccine reducing fertility, but further research will be needed to confirm this. Boarding students were more willing to be vaccinated; we speculate that they may be more influenced by teachers than by their parents, given findings that parental vaccine hesitancy, refusals, delays, and negative attitudes may influence adolescent vaccine hesitancy or deferral [35]. We were surprised that students in Beijing were less willing to be vaccinated, but this may also be parentally influenced. In the Tongzhou District of Beijing, only about 60% of parents chose to vaccinate themselves and their children; this is lower than their willingness to receive influenza or HPV vaccines [36]. In contrast, 79.2% of residents living in Guangzhou were willing to be vaccinated [37].

Students who have been vaccinated against influenza were more willing to receive a COVID-19 vaccine. Knowledge of influenza immunization benefits are relevant to the awareness of similar advantages in preventing COVID-19 and other respiratory viral infections [38]. Experience with the influenza vaccination was reported to be associated with higher COVID-19 vaccine acceptance [39]. The promulgation of the influenza vaccination may increase the possibility of Chinese adults to accept the COVID-19 vaccination [40], and our findings suggest that influenza vaccine efforts may also work to scale-up COVID-19 vaccination willingness among students.

Vaccine hesitancy has been suggested as one of the top 10 threats to global health in 2019 [41]. We found that vaccine hesitancy was associated with a low awareness of SARS-CoV-2, while vaccine resistance was associated with a lower perceived severity of COVID-19 [19]. Self-perception of being at a high risk of developing a serious COVID-19 infection has become a positive predictor of future vaccination [42]. HBM theory supports the premise that education about the content in advance, addressing of prior symptoms or concerns, and receiving the correct information from the mass media are cues which can influence behavior and help to prevent COVID-19 via an increase in vaccination uptake. We found that students who did not realize the magnitude of the COVID-19 pandemic’s impact were more likely to be resistant or hesitant to a vaccine. According to a WHO/UNICEF joint report using 2015–2017 data, the principal reasons for vaccine hesitancy were a lack of knowledge and awareness of vaccination and a lack of understanding of its importance [43]. In addition, people who are vaccine-resistant have been exposed to significantly less information about COVID-19 from the public media compared to those willing to receive a vaccine [32].

In our study, students overwhelmingly cited the safety (93.0%) and effectiveness (82.9%) of a vaccine as major considerations affecting their decision. Vaccine hesitancy was related to concerns about vaccine effectiveness, lower perceived convenience, higher price, and lack of doctors’ recommendation. According to China’s current vaccine immunization plan, all people eligible for a COVID-19 vaccine will receive it for free, so this is information that must be conveyed to youth. Vaccination inconvenience hinders people from promptly vaccinating, a remediable challenge [40].

Vaccine safety is essential to “first do no harm” and to maintain public trust in vaccines. Each COVID-19 vaccine must be demonstrated to be safe for adolescents and then for younger children at appropriate doses, at which time the authorization from regulatory agencies will be broadened as per the data-based evidence. COVID-19 vaccines are likely to be even more effective and safe than other childhood vaccines [44]. High COVID-19 vaccine effectiveness and safety are playing a pivotal role in chipping away at vaccine hesitancy over time [45]. Vaccine hesitancy was initially related to public perceptions of ineffective vaccines and safety concerns [46]. We found that, in Chinese youth, doctors’ recommendation may decrease vaccine hesitancy [40]. People who are vaccine-resistant report lower levels of trust in information released by their doctors, other health care professionals, and/or government agencies [32]. It is plausible that excellent, consistent, persistent, and effective public communications can help improve vaccination uptake regardless of the setting and causes of vaccine hesitancy [8].

Our study’s strengths include our provision of baseline information for continuously monitoring teenagers’ acceptance of the COVID-19 vaccine in China. Given what we discovered to be the major concerns of young people, a blueprint for education is suggested for adolescents. Our four cities provide geographical diversity throughout China, improving the generalizability potential of our findings. We acknowledge several limitations of our study. First, though the 9153 students represent a considerable overall sample, a study limitation includes the low representation of students in Anqing City. A second limitation is that recall bias can be influential in a cross-sectional survey, such as with remembering one’s influenza vaccine experience—though forgetting about an injection is presumably less common in teenagers than for younger children. Third, we did not investigate vaccine hesitancy across different types of COVID-19 vaccines and the psychology of vaccine hesitancy and resistant individuals (such as personality, altruism, or domains from mental health scales, such as Revised Children’s Anxiety and Depression Scales (RCADS)) to further understand the reason for vaccine hesitancy. In addition, misinformation can lead to hesitancy.

As of February 2022, over 3 billion doses of the COVID-19 vaccine have been administered, according to the report of National Health Commission of the People’s Republic of China [47]. Our study provides an early glance into the willingness and potential hesitancy to receive of a COVID-19 vaccine among students in grades 7–12 in urban China. With the current deployment of vaccines to teenagers, these data can help with baseline information to monitor changes in vaccine acceptability among adolescents. Our results highlight that vaccine hesitancy is associated with a limited understanding of SARS-CoV-2 transmission dynamics and insufficient risk awareness [48,49]. We recommend that schools, community leaders, health care providers, governments, and parents improve information, education, and communication of the comparative benefits of a COVID-19 vaccine among adolescents.

## 5. Conclusions

In conclusion, vaccine hesitancy in Chinese adolescents was associated with limited health literacy and lower risk perception. More youth-friendly messages regarding COVID-19, as well as the safety and effectiveness of the immunization against misinformation, may help address vaccine hesitancy. Our findings can inform public health campaigns to boost the immunization rate.

## Figures and Tables

**Figure 1 vaccines-10-00452-f001:**
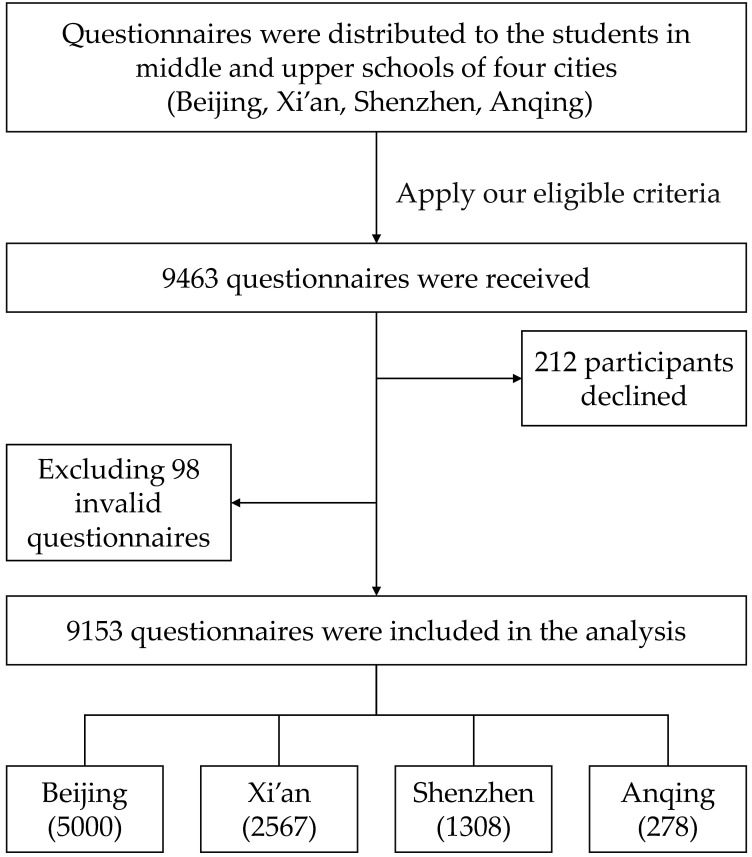
Recruitment flowchart for the cross-sectional survey among school students in grades 7–12 in four cities in China.

**Figure 2 vaccines-10-00452-f002:**
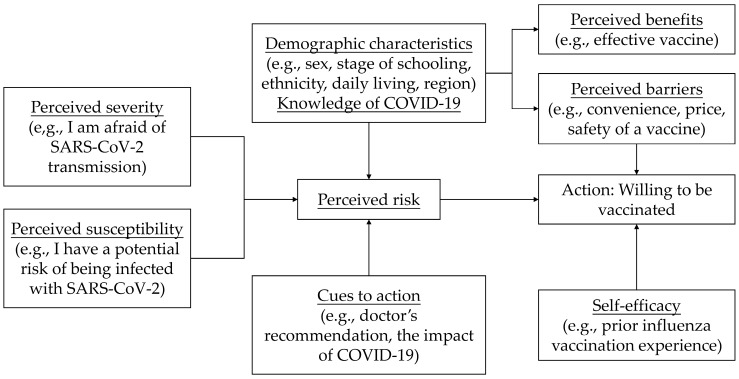
Components of theory of a health belief model as applied to COVID-19 vaccine willingness, hesitancy, or resistance in Chinese adolescents.

**Table 1 vaccines-10-00452-t001:** Demographic characteristics and COVID-19 vaccination concerns of school students in grades 7–12 in four cities in China (N = 9153).

Demographic Characteristics	*n* (%)
Sex	
Boy	4575 (50.0)
Girl	4578 (50.0)
Daily living	
Boarding at school	1822 (19.9)
Not boarding	7331 (80.1)
Ethnicity	
Han	8618 (94.2)
Minorities	535 (5.8)
Stage of schooling	
Junior high (middle) school grades 7–9	5708 (62.4)
Senior high (upper) school grades 10–12	3445 (37.6)
Region	
Beijing	5000 (54.6)
Anqing, Anhui province	278 (3.0)
Shenzhen, Guangdong province	1308 (14.3)
Xi’an, Shaanxi province	2567 (28.1)
COVID-19 concerns	
I am afraid of SARS-CoV-2 transmission	
Agree	7248 (79.2)
Not sure	852 (9.3)
Disagree	1053 (11.5)
I have a potential risk of being infected with SARS-CoV-2	
Agree	3550 (38.8)
Not sure	2536 (27.7)
Disagree	3067 (33.5)
The COVID-19 has made me pay more attention to the influence of infectious diseases in my life, influencing me to get/want an influenza vaccine	
Agree	6412 (70.0)
Not sure	2059 (22.5)
Disagree	682 (7.5)
COVID-19 vaccination concerns	
I will get the COVID-19 vaccine	
Agree	5497 (60.0)
Not sure	2891 (31.6)
Disagree	765 (8.4)
Major concerns that affect my COVID-19 vaccination decision	
Safety	8516 (93.0)
Effectiveness	7587 (82.9)
Price	3520 (38.5)
Convenience	3448 (37.7)
Doctors’ recommendation	3116 (34.0)
Influenza vaccination	
Did you get influenza vaccination in 2020	
Yes	3494 (38.2)
No	5659 (61.8)
I have experience with a prior influenza vaccination	
Agree	5240 (57.2)
Not sure	1992 (21.8)
Disagree	1921 (21.0)

Note: The statement, “Major concerns that affect my COVID-19 vaccination decision”, has multiple responses, including safety, effectiveness, price, convenience, and a doctor’s recommendation.

**Table 2 vaccines-10-00452-t002:** Demographic characteristics and COVID-19 vaccination concerns of school students in grades 7–12 in four cities in China, by willingness to getting the vaccine (N = 9153).

Characteristics	Resist to Be Vaccinated (N = 765), *n* (%)	Hesitate to Be Vaccinated (N = 2891), *n* (%)	Willing to Be Vaccinated (N = 5497), *n* (%)	*p*
Demographic characteristics				
Sex				<0.001
Boy	392 (8.6)	1356 (29.6)	2827 (61.8)	
Girl	373 (8.1)	1535 (33.5)	2670 (58.3)	
Daily living				0.015
Boarding at school	137 (7.5)	537 (29.5)	1148 (63.0)	
Not boarding	628 (8.6)	2354 (32.1)	4349 (59.3)	
Ethnicity				0.836
Han	722 (8.4)	2716 (31.5)	5180 (60.1)	
Minorities	43 (8.0)	175 (32.7)	317 (59.3)	
Stage of schooling				0.012
Junior high (middle) school grades 7–9	512 (9.0)	1766 (30.9)	3430 (60.1)	
Senior high (upper) school grades 10–12	253 (7.3)	1125 (32.7)	2067 (60.0)	
Region				<0.001
Beijing	481 (9.6)	1680 (33.6)	2839 (56.8)	
Anqing, Anhui province	14 (5.0)	105 (37.8)	159 (57.2)	
Shenzhen, Guangdong province	63 (4.8)	356 (27.2)	889 (68.0)	
Xi’an, Shaanxi province	207 (8.1)	750 (29.2)	1610 (62.7)	
COVID-19 concerns				
I am afraid of SARS-CoV-2 transmission				<0.001
Agree	507 (7.0)	2010 (27.7)	4731 (65.3)	
Not sure	39 (4.6)	595 (69.8)	218 (25.6)	
Disagree	219 (20.8)	286 (27.2)	548 (52.0)	
I have a potential risk of being infected with SARS-CoV-2				<0.001
Agree	166 (4.7)	731 (20.6)	2653 (74.7)	
Not sure	117 (4.6)	1215 (47.9)	1204 (47.5)	
Disagree	482 (15.7)	945 (30.8)	1640 (53.5)	
The COVID-19 has made me pay more attention to the influence of infectious diseases in my life, influencing me to get an influenza vaccine				<0.001
Agree	245 (3.8)	1169 (18.2)	4998 (77.9)	
Not sure	158 (7.7)	1570 (76.3)	331 (16.1)	
Disagree	362 (53.1)	152 (22.3)	168 (24.6)	
COVID-19 vaccination concerns				
Major concerns that affect my COVID-19 vaccination decision				<0.001
Safety	680 (8.0)	2667 (31.3)	5169 (60.7)	
Effectiveness	571 (7.5)	2326 (30.7)	4690 (61.8)	
Price	213 (6.1)	1100 (31.3)	2207 (62.7)	
Convenience	208 (6.0)	999 (29.0)	2241 (65.0)	
Doctors’ recommendation	209 (6.7)	908 (29.1)	1999 (64.2)	
Influenza vaccination				
Did you get influenza vaccination in 2020				<0.001
Yes	242 (6.9)	972 (27.8)	2280 (65.3)	
No	523 (9.2)	1919 (33.9)	3217 (56.8)	
I have experience with a prior influenza vaccination				<0.001
Agree	353 (6.7)	1336 (25.5)	3551 (67.8)	
Not sure	143 (7.2)	890 (44.7)	959 (48.1)	
Disagree	269 (14.0)	665 (34.6)	987 (51.4)	

**Table 3 vaccines-10-00452-t003:** Factors associated with the willingness of receiving a COVID-19 vaccine among school students in grades 7–12 in four cities in China—multinomial logistic regression.

Factors	Intend to Be Vaccinated: Vaccine Resistance vs. Willing		Intend to Be Vaccinated: Vaccine Hesitancy vs. Willing	
	OR (95% CI)	*p*	OR (95% CI)	*p*
Sex				
Boy	Ref		Ref	
Girl	1.04 (0.87–1.24)	0.692	1.21 (1.09–1.36)	<0.001
Daily living				
Boarding at school	0.78 (0.61–1.00)	0.052	0.79 (0.68–0.92)	0.003
Not boarding	Ref		Ref	
Stage of schooling				
Junior high (middle) school grades 7–9	0.99 (0.80–1.22)	0.913	0.98 (0.86–1.11)	0.725
Senior high (upper) school grades 10–12	Ref		Ref	
Region				
Beijing	1.81 (1.44–2.28)	<0.001	1.62 (1.40–1.88)	<0.001
Anqing, Anhui province	0.65 (0.34–1.25)	0.200	1.23 (0.89–1.71)	0.213
Shenzhen, Guangdong province	0.72 (0.51–1.01)	0.056	0.90 (0.75–1.09)	0.282
Xi’an, Shaanxi province	Ref		Ref	
I am afraid of SARS-CoV-2 transmission				
Agree	Ref		Ref	
Not sure	0.87 (0.58–1.29)	0.487	2.09 (1.70–2.57)	<0.001
Disagree	1.35 (1.07–1.71)	0.013	0.93 (0.78–1.12)	0.458
I have a potential risk of being infected with SARS-CoV-2				
Agree	Ref		Ref	
Not sure	1.43 (1.08–1.89)	0.012	1.93 (1.67–2.22)	<0.001
Disagree	3.57 (2.86–4.46)	<0.001	1.72 (1.50–1.97)	<0.001
The COVID-19 has made me pay more attention to the influence of infectious diseases in my life, influencing me to get an influenza vaccine				
Agree	Ref		Ref	
Not sure	8.03 (6.30–10.22)	<0.001	14.15 (12.27–16.31)	<0.001
Disagree	27.68 (21.81–35.13)	<0.001	3.24 (2.56–4.11)	<0.001
Major concerns that affect my vaccination				
Safety	0.80 (0.58–1.10)	0.167	1.20 (0.95–1.50)	0.130
Effectiveness	0.66 (0.52–0.83)	<0.001	0.84 (0.72–0.98)	0.027
Price	0.86 (0.69–1.07)	0.169	1.20 (1.05–1.36)	0.007
Convenience	0.81 (0.64–1.01)	0.064	0.84 (0.73–0.96)	0.011
Doctors’ recommendation	0.93 (0.75–1.15)	0.496	0.86 (0.76–0.98)	0.025
Did you get influenza vaccination in 2020				
Yes	0.88 (0.72–1.09)	0.240	0.89 (0.78–1.01)	0.063
No	Ref		Ref	
I have experience with a prior influenza vaccination				
Agree	Ref		Ref	
Not sure	1.30 (1.01–1.67)	0.038	1.44 (1.24–1.67)	<0.001
Disagree	1.57 (1.25–1.98)	<0.001	1.33 (1.14–1.55)	<0.001

## Data Availability

The data that support the findings of this study are not publicly available but are available from the corresponding author on reasonable request.

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
