# Peer review of "COVID-19 Vaccine Hesitancy among Adolescents: Cross-Sectional School Survey in Four Chinese Cities Prior to Vaccine Availability"

_vaccines, 2022, doi:10.3390/vaccines10030452_

Round 1
Reviewer 1 Report
The manuscript provides an interesting insight into COVID-19 vaccine hesitancy among adolescents in China. The topic is timely and the population under investigation is critical to reach an adequate immunization rate. A few suggestions to improve the manuscript.
Introduction
This section needs to be updated. Line 45, for example, refers to data until June 2021. Line 50 and 51 as well.
Line 65. Previous studies have investigated university students also. Since they are close to adolescents in terms of age and maybe also perception and experience, this should be mentioned. Citations worth including are doi: 10.1371/journal.pone.0255447, doi: 10.3390/vaccines9111292, and doi: 10.3390/vaccines10010105.
Methods
Figure 1 should be described in the results section. Please also correct the direction of the arrows for people who declined the survey or questionnaires that were excluded.
It’s unclear how you administered the survey. You wrote online questionnaires, but how did you calculate the response rate (i.e., how many students were approached?).
Discussion
What is the immunization rate among adolescents in China as of February 2022? This data should be mentioned and discussed in light of these findings.
Reviewer 2 Report
Journal: Vaccines
Manuscript ID: 1605527
Manuscript Type: Original Article
Date: 2022.02.19
Manuscript title: COVID-19 Vaccine Hesitancy among Adolescents: Cross-sectional School Survey in Four Chinese Cities Prior to Vaccine Availability
Comments for the authors:
Abstract
- Line 23. Please, specify between brackets participants’ M and SD for age as well as the percentage of males/females.
Introduction
- Lines 65-67. The sentence “Previous studies on the willingness of COVID-19 vaccination included adults and little is known about adolescents’ acceptance of a COVID-19 vaccine or factors affecting acceptance” is only partially correct. Although less numerous, some previous studies about vaccine hesitancy on adolescents have already been published and should be cited at this point. See for example the references below:
Willis et al., 2021. COVID-19 vaccine hesitancy among youth. Human Vaccines & Immunotherapeutics, DOI: 10.1080/21645515.2021.1989923
Fazel et al., 2021. Willingness of children and adolescents to have a COVID-19 vaccination: Results of a large whole schools survey in England. EClinicalMedicine, DOI: 10.1016/j.eclinm.2021.101144
Afifi et al., 2021. Older adolescents and young adults willingness to receive the COVID-19 vaccine: Implications for informing public health strategies. Vaccine, DOI: 10.1016/j.vaccine.2021.05.026
Brandt et al., 2021. National Study of Youth Opinions on Vaccination for COVID-19 in the U.S. Journal of Adolescent Health, DOI: 10.1016/j.jadohealth.2021.02.013
Wong et al., 2022. Adolescents’ attitudes to the COVID-19 vaccination. Vaccine, DOI: 10.1016/j.vaccine.2022.01.010
Middlemn et al., 2022. Vaccine Hesitancy in the Time of COVID-19: Attitudes and Intentions of Teens and Parents Regarding the COVID-19 Vaccine. Vaccines, DOI: 10.3390/vaccines100100004
- Line 68. Please specify when this vaccination plan has been initiated.
- What this study adds to the previous literature should be clearly specified.
Materials and Methods
- Line 92. Who were these experts? Clinicians, psychologists, researchers?
- Lines 92-93. How many participants have been involved in these pilot surveys? What were their characteristics?
- Line 97. Please explain the reason why vocational schools were not included.
- Lines 119-121. The authors state “We defined being unsure as to whether to accept vaccination as vaccine hesitancy. Unwillingness to accept vaccination was defined as vaccine resistance” (lines 104-105). However, at this point they use “unwilling or resistant” and “I am not sure or hesitant”. Please, be consistent.
- Lines 122-132. The description of the measures lacks in some points. In my opinion, it is not clear how each variable has been evaluated. How many items for each variable? Were the questions adapted from previous validated measures or were specifically created for this study? Figure 2 could support the description of the measures, but it cannot replace it. In this version it is not possible to completely evaluated if variables have been measured in a proper way.
- Lines 123-124. “Concerns about coronavirus and its vaccination” are not cited in figure 2.
- Lines 134-136. This sentence should be rephrased, as descriptive statistics do not evaluate associations between variables.
- Line 136. Since the study has a cross-sectional design, no conclusions about causality can be drawn (thus, the word “predictors” is not suitable).
Results
Lines 142-148. Information that is already in Figure 1 could be removed from the text in order to avoid repetitions.
Lines 152-164. As above, data that are already in Table 1 could be removed from the text in order to avoid repetitions.
Lines 175-205. Please report also the likelihood ratio test of the final models against the null model (with intercept only), as well as R2 values (Cox & Snell and Nagelkerke).
Discussion
- Lines 209-213. In my opinion, the results of the study (i.e. the percentage of acceptant, hesitant and resistant) should be compared also with other previous studies on adolescents.
- Lines 220-224. This hypothesis should be clarified. Is the percentage of parents who chose to vaccine themselves lower for those in Beijing than in other regions?
- Lines 225-226. Being vaccinated against influenza is not related to vaccine hesitancy in multinomial logistic regression (Table 3). Although, having experience with a prior influenza vaccination is related to vaccine hesitancy (Table 3). Did the authors have some hypotheses which could explain this?
- Previous studies (e.g. Salerno et al., 2021 – Vaccines) showed that the rate of vaccine hesitancy differs with respect to the vaccine typology (mRNA, viral vector). The present study does not differentiate between the different types of COVID-19 vaccines. This aspect should be discussed in the Limitations section.
- In order to explain which factors may influence people willingness to be vaccinated, the authors refer to demographic characteristics, cognitive factors (e.g. perceived susceptibility, perceived severity, …) and “objective” data (e.g. price of the vaccine). However, it is not mentioned that also emotional and individual psychological variables can influence participant’s vaccine hesitancy (e.g. Murphy et al. 2021 – Nat. Commun). Also this point should be discussed in the limitations section.
Figure 1
- Please, specify why 98 questionnaires have been considered as “invalid”.
- “totally” and “finally” should be removed.
Table 1
- I suggest specifying in note that participants could score multiple responses for the question “Major concerns that affect my COVID-19 vaccination decision”.
Table 2
- As stated above, the authors “defined being unsure as to whether to accept vaccination as vaccine hesitancy. Unwillingness to accept vaccination was defined as vaccine resistance” (lines 104-105). However, in this table they use “unwilling to be vaccinated”. Please, be consistent.
Minor points
- Please, check for typo (e.g. in the abstract, the “;” before “Of 9153 students […]”.
Reviewer 3 Report
This paper is formulated from the actual situation about vaccination in health crisis produced around the COVID-19 pandemic.
This is a topic of great interest. However, questions about COVID and vaccination strategies are changing rapidly. In this sense, the analysis of the study sample was carried out in December 2020 and the vaccination strategies in the child and adolescent population have changed in the different countries. This fact supposes a problem in the contextualization of the paper that is submitted to evaluation.
It would be interesting if the authors included in the paper information on the vaccination strategy followed in China for the population and, also, for minors.
The paper must also be conveniently contextualized. It has been carried out in four Chinese cities and the reasons why these cities have been chosen should be explained to the reader.
The abstract should be revised, and its structure improved. The terms resistant/willing are used in a similar sense in the text (Line 24). I believe that they are terms that should be treated separately and not together as is done at paper.
In this manuscript the objectives, both general and specific, of the study presented must be clearly specified.
Line 105: For the authors ‘Unwillingness to accept vaccination was defined as vaccine resistance’. In my opinion, unwillingness to accept vaccination is not always similar to vaccine resistance.
Line 147: The study sample includes youth from 12.0-17.5 years. I don't understand this age distribution. What does 17.5 years, 17 years and 6 months mean?
Are there differences in the responses to the questionnaire in relation to the age groups of the young people?
Tables 1,2 and 3 should be presented in a more appropriate way.
In the discussion section, ideas are formulated but lacking in concreteness and many of them are very general. Some of the explanations in the discussion lack a consistent explanation. For example, in lines 220 and 221 the authors show their surprise at the results obtained in the Beijing students. The explanation is that this may also be parentally influenced. Why in Beijing and not in the other cities?
It would be interesting to include some consistent and solid conclusions.
The list of references must be reviewed and cited correctly.
Round 2
Reviewer 2 Report
The revised manuscript has addressed some issues raised by the reviewers. However, other minor revisions are outlined below:
- Line 21. Please, specify that 14.2 is the mean (e.g. “[…] students with a mean age of 14.2 […]”. I still suggest to report the SD for age.
- Lines 29-30. In order to be consistent across the abstract, please report values also for “postulating a highly effective vaccine”.
- Lines 32-33. In order to be consistent across the abstract, report either one or two decimal places.
- Lines 33-34. I suggest to link this sentence to the rest of the text, specifying that it is a summary of the results that are described above (e.g. “In conclusion, the results of the study showed that vaccine hesitancy […]”, or “Globally, our findings show that vaccine hesitancy […]”).
- Lines 67-68. I suggest to briefly report also the findings from these previous studies on adolescents’ vaccine hesitancy. I think it is important in order to understand what this study adds to the previous literature on this topic.
- Lines 105-115. I suggest moving this description to the “measurement” section.
- Lines 112-114. The authors replace only to the first part of my question (i.e. “how many participants have been involved in these pilot surveys”), but not to the second (i.e. “What were their characteristics?”). Please, specify at least participants’ M and SD for age as well as the percentage of males/females.
- Line 115. The sentence “There were 16 variables has been evaluated” should be rephrased.
- Line 140. What about “school location”? (i.e. no information about this variable is provided in table 1).
- Lines 283-284. The sentence “the psychology of vaccine hesitancy and resistant individuals” should be better clarified. In other words, the authors should provide some examples of psychological variables that have been related to vaccine hesitancy.
- Lines 21 and 79. Check for typo (e.g. line 21: there is a space missing before the word “student”).
- Figure 1. Following lines 89-90, the sentence “Questionnaires were distributed to the students in middle schools” should be “[…] in middle and upper school”.
- More broadly, in this revised version of the manuscript the authors have greatly improved the description of the measurements. However, I think that the readability of the work could be further improved if attention will be paid to greater consistency of terminology between Figure 2 and Tables 1-3. I wonder why not use also in tables the same labels used in figure 2, rather than define the variables in another way or indicate only the text of the items. For example, why include in the title "COVID-19 concerns" two questions (i.e. "I am afraid of coronavirus transmission" and "I have a risk of being infected by coronavirus") instead of indicate them as "perceived severity" and "perceived susceptibility" (in line with Figure 2 and the HBM model)?
Reviewer 3 Report
The article has been substantially improved.
The authors have considered the reviewers' suggestions and the different sections of the paper have been improved and clarified, as far as I'm concerned it can be published.
